# Patient-perceived factors influencing physical activity sensor use in stroke prevention and rehabilitation: A thematic synthesis protocol

**Paul T. Harris**[1]*, **Ingrid Maine**[1,2]

**1** Department of Medical Education, The University of Melbourne, Parkville, Victoria, Australia, **2** Geriatric Evaluation and Management, Western Health, St Albans, Victoria, Australia

* paulharris@alumni.unimelb.edu.au

**Data Availability Statement:** Data underlying the results of the proposed study are publicly available

## Abstract

### Introduction

While the putative benefits of "fitness trackers" continue to fuel a booming consumer market, results of device-based clinical interventions remain remarkably mixed. This study will explore factors influencing wearable physical activity (PA) sensor use in the context of stroke prevention and rehabilitation for older adults. The findings of this thematic synthesis will provide insights into factors influencing the use of PA sensors in stroke which may inform more effective device-based interventions.

### Methods and analysis

Thematic synthesis as a formal method described by Thomas and Arden can be used within a systematic review to synthesize primary qualitative research. Accordingly, the proposed study will systematically search bibliographic databases for relevant peer-reviewed papers and synthesize coded thematic data within included papers. The quality of papers will be assessed using the JBI Critical Appraisal Checklist for Qualitative Research. Patterns in the text will be coded, preliminary data visualised, and higher-level analytical themes discerned to explain factors influencing the use of PA sensors in older stroke patients.

### Discussion

This study does not require ethics approval. Results are expected to be available by June 2024. Data from the thematic synthesis will provide insights into barriers and facilitators influencing the use of wearable PA sensors in stroke and older adults at risk, and implications these factors have for the design of effective device-based interventions.

### Trial registration

**Systematic review registration**: PROSPERO registration number: CRD42020211472. https://www.crd.york.ac.uk/prospero/display_record.php?ID=CRD42020211472.

from the figshare repository (https://doi.org/10.6084/m9.figshare.25573209).

**Funding:** The author(s) received no specific funding for this work.

**Competing interests:** The authors have declared that no competing interests exist.

## Introduction

Stroke incidence has markedly declined in high-income countries over the last four decades, arguably at least in part due to improved systems of care and secondary prevention strategies. Even still, stroke remains a leading cause of disability and death in Australia, and the number of people living with the effects of stroke is projected to double by 2050 [1].

The risk factors for stroke are a combination of non-modifiable attributes such as age and genetics; medically modifiable factors addressed through pharmaceutical or surgical intervention; and those that may be altered to some degree through lifestyle or behavior change [2]. Age is the most profound non-modifiable risk factor for incident stroke, doubling in both men and women every successive decade of life from age 55 years [3]. While in Australia stroke prevalence is highest for older (65 years or more) and elderly adults (85 years and over), rates begin to rise from 50 and older [4].

One way to manage underlying risk factors such as hypertension and reduce the likelihood of first-time or recurrent stroke is through increased PA. Australian guidelines recommend 30 minutes moderate exercise daily for older adults, which may include brisk walking, gardening, swimming, and other activities [5]. Self-report data for Australian adults suggests only 35% do at least 150 minutes exercise over five or more sessions, 37% less than 150 minutes, and 28% reported no regular exercise. In fact, 56% of Australian adults are inactive or have low levels of physical activity [6].

Over the last two decades wearable PA sensors have evolved substantially, recording step counts, exercise duration, cadence or intensity, cardiovascular fitness (heart rate, VO2 Max), and estimated calorie consumption. Commercially available, consumer-grade devices include fitness bands and smartwatches with integrated accelerometers and gyroscopes that detect motion, and photoplethysmogram measurements of heart rate and blood pressure changes. Smartwatches and wearable sensors have already been trialed in clinical settings for long-term weight loss, treatment compliance (e.g., exercise, diet, smoking), and task prompting in memory impaired stroke patients [7–10]. In some stroke clinics wearable medical-grade sensors have been used to measure walking activities [11].

While the use of consumer-grade devices has been associated with increased PA, qualitative research on efficacy in stroke prevention and rehabilitation remains remarkably sparse. Surprising given reported associations between PA and reduced stroke risk, and evidence suggesting wearable sensors may effectively promote increased PA [12]. Healthcare interventions necessarily involve individuals with the freedom to act in ways that can affect intervention outcomes. Anticipating and retrospectively unravelling these factors can be a complex undertaking. Qualitative research techniques are well suited to the study of human experience factors, revealing aspects that may not be measurable or otherwise evident quantitatively.

The aim of this study is to identify factors (barriers and facilitators) influencing the use of wearable PA sensors specifically in stroke patients and older adults at risk; and explore implications these have for effective PA sensor-based interventions in this population.

## Methods and analysis

### Study design

This protocol has been informed by and designed in accordance with the Preferred Reporting Items for Systematic Review and Meta-Analysis Protocols (PRISMA-P) statement [13] and checklist [14] (S1 Checklist). The protocol was registered with PROSPERO (ID: CRD42020211472) with changes to the published record to be reported. Research questions and search criteria were developed using a version of the Population, Intervention,

Comparison and Outcome (PICO) tool adapted for qualitative studies incorporating Problem or Population of interest and Context (PICo) elements [15]. Context here refers to societal, temporal, and condition characteristics influencing the efficacy of PA sensor-based healthcare interventions. Due to the paucity of stroke-specific qualitative data, other patient groups with chronic conditions (risk factors) and deficits (motor and cognitive) like those found in stroke will also be included. While "older adult" more often refers to individuals 65 years and over, for the purpose of this study it is defined more broadly from age 50 years when the risk of stroke begins to increase.

**Eligibility criteria.** Papers will be included if: peer-reviewed and published in English from 1 January 2010 to 31 December 2023; primary focus on wearable physical activity sensor use; qualitative design, or significant qualitative component (e.g., mixed methods); population are older adults (50 years and over), stroke or with similar risk factors and deficits. Papers will be excluded if: they are literature reviews and descriptive articles, conference proceedings, grey literature, protocols, or non-human studies; there is no focus on user experience; quantitative design; population are under 50 years, or with acute psychiatric illness.

**Search strategy.** A sensitive search strategy combining medical subject headings (MeSH) and keywords will be collaboratively developed by the review team (IM, PH) and a research librarian. This will be optimized first for MEDLINE (Box 1), then adapted for each target database. Search terms will include wearable electronic devices; monitoring, physiologic; wearable; accelerometry; physical activity; exercise; qualitative. Result subsets will be combined using standard Boolean operators. Filters for study design and method will be avoided so that any research with a qualitative component will be captured including mixed method, randomized controlled trials, observational and cross-sectional studies with an embedded qualitative component. Only English-language studies published for a decade from 1 January 2010 to 31 December 2023 will be included. Duplicate papers from the searches will be removed. Bibliographies from selected articles will be manually scanned for additional references.

---

Box 1. Systematic review high-level search terms

**DATABASES:** EMBASE, CINAHL, Cochrane, MEDLINE, PsycINFO, Scopus

(Wearable Electronic Devices/ OR wearable*.mp. OR Monitoring, Ambulatory/

OR Monitoring, Physiologic/)

AND (physical activity.mp. OR Exercise/)

AND (physical activity.mp. OR Exercise/)

AND limit 5 to (Human, English language AND yr = "2010–2023")

---

**Screening procedure.** Searches will be independently performed by two reviewers (IM, PH), initially scanning publication titles and abstracts against the eligibility criteria (Table 1). A pilot test, screening example articles against the inclusion and exclusion criteria, will promote mutual understanding between reviewers. A simple agreement rate for the screening process will be calculated, reflecting the proportion of cases where both reviewers agree on the inclusion or exclusion of a paper. A high simple agreement rate would be indicative of more consistent understanding between the reviewers. Candidate citations identified through bibliographic searches will be merged in a shared, cloud-based spreadsheet. Any disagreements will

**Table 1. Inclusion and exclusion criteria for systematic study selection.**

| Included | Excluded |
| --- | --- |
| Peer-reviewed articles, published in English | Literature reviews and descriptive articles, conference proceedings, grey literature, protocols, non-human studies |
| Wearable physical activity sensor use | No focus on user experience |
| Qualitative, or with significant qualitative component (e.g., mixed methods) | Quantitative studies |
| Older adult (50 years +), stroke and similar risk factors or deficits | Under 50 years, acute psychiatric illness |
| Published from 1 January 2010 to 31 December 2023 | |

be resolved through discussion in the first instance, or through a third reviewer as required. Search result set and screening counts will be summarized in a PRISMA flow diagram [13].

**Data extraction.** Off-topic citations not relevant to the review will be eliminated along with duplicates before full text candidate papers are evaluated in Covidence (www.covidence.org). A data extraction form will be used to capture study characteristics including publication year, country, number of participants, age range, research design, intervention characteristics (i.e., device type) and collection methods. These attributes will also be recorded in an NVivo classification sheet for analytical purposes. Other potential citations will be harvested from the reference lists of selected papers.

**Quality assessment.** Reporting quality is essential for the appraisal of existing research. Candidate papers will be assessed for congruity of perspective, research questions/objectives, methodology, bias, participant voice, ethical standard, analysis and conclusions, and overall quality using the JBI Critical Appraisal Checklist for Qualitative Research [16]. Papers will also be evaluated for analytical richness, reflecting how deeply qualitative data is explored and interpreted with respect to the aims of the proposed study [17].

## Data synthesis

Three stages have been described for the thematic synthesis methodology [18]. In the first stage, inductive coding of included papers will be performed independently by two reviewers examining the text line-by-line, identifying salient sections, and discussing coded or labelled sections as a team. Codes are the smallest unit of meaning, encapsulating a concept or patterned response within the text relevant to the research question(s) [18]. It is anticipated that the reviewers will agree upon a coding frame based on preliminary codes, revising these through discussion and consensus. The coding frame will then be applied to the findings of all selected papers. At this stage, coding will be checked to ensure consistency of interpretation (referred to in grounded theory as "axial coding"), and that themes remain connected to and understood within the context of the source studies [18].

In the next stage the review team will group related codes into broader "descriptive themes" that summarize concepts (codes) across studies, while not going beyond what they represent in the source text. Summaries will be drafted, discussed, and refined, referring back to the source. Finally, descriptive themes will be analysed with respect to the research questions and objectives. Links will be mapped between descriptive themes to generate higher-level analytical constructs [18]. The review team will conduct this interpretive process collaboratively to develop a coherent picture from the evidence base in relation to the research questions [19].

A set of hierarchically organised a priori codes (Fig 1) based on the Person-Environment-Occupation-Performance (PEOP) and Technology Acceptance Model (TAM) will provide a

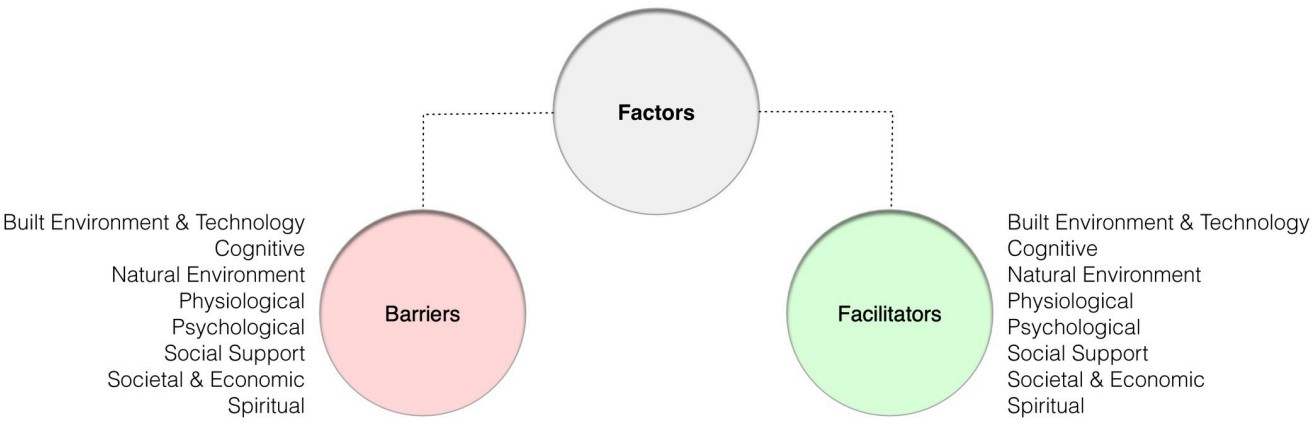

**Fig 1. Preliminary node structure.**

provisional structure for inductive coding [20,21]. The PEOP, a client-centred model developed in the mid-1980s, defines intrinsic factors (psychological, cognitive, neurobehavioural, physiological) and extrinsic or environmental factors (social support, societal policies and attitudes, natural and built environments, cultural norms and values) that influence the performance of occupations or activities. More specifically, these factors may be applied to the efficacy of wearable PA sensors for the promotion of physical activity in stroke prevention and rehabilitation [20]. The TAM, which also emerged in the 1980s, was adapted from behavioural theory specifically to understand the use of emerging technologies. This model defined constructs such as perceived usefulness and perceived ease of use [21].

Full text copies of included papers will be imported into NVivo (QSR International, Melbourne, Australia) for coding. Node redundancy will be used to simplify coding and synthesis, with codes and themes mirrored under top-level facilitator and barrier nodes. Studies will be examined line-by-line in their entirety, independently by the reviewers. Commentary on the existing literature, results, discussion, and conclusion sections of included studies will be appraised, with codes assigned to salient phrases, sentences, or paragraphs. Direct quotes from participants are relevant in thematic analysis but not synthesis, so these will not be coded. Themes explicitly relevant to PA sensor use, patently applicable to stroke will be coded. Those of more general relevance to physical activity or wellbeing with no direct bearing on sensor use will be ignored.

The coding structure will necessarily reflect an artificial barrier/facilitator binary, defining themes in obverse terms. Valence will be used to differentiate subjective barrier and facilitator themes, with negative valence characterizing inhibitory factors (barriers), and positive valence enabling factors (facilitators). The reviewers will work up from the source texts in an iterative process described earlier, recording individual observations and consensus decisions in reflexive notes, until all themes relevant to the research questions have been discerned. Codes will then be reviewed for similarities and differences, and grouped under unifying themes, based on similarity of meaning or user experience (Fig 2). Overarching descriptive themes identified provisionally and through inductive coding may also be used to infer the top-level node (barrier or facilitator) where this is not explicitly evident.

Coded region frequencies (number of times themes are coded) and referencing study counts (number of studies reported within) will be analyzed and reported. While coding "density" metrics will be indicative only of theme recurrence for included papers, measures such as these are well established in qualitative research and used much like matrix-coding queries

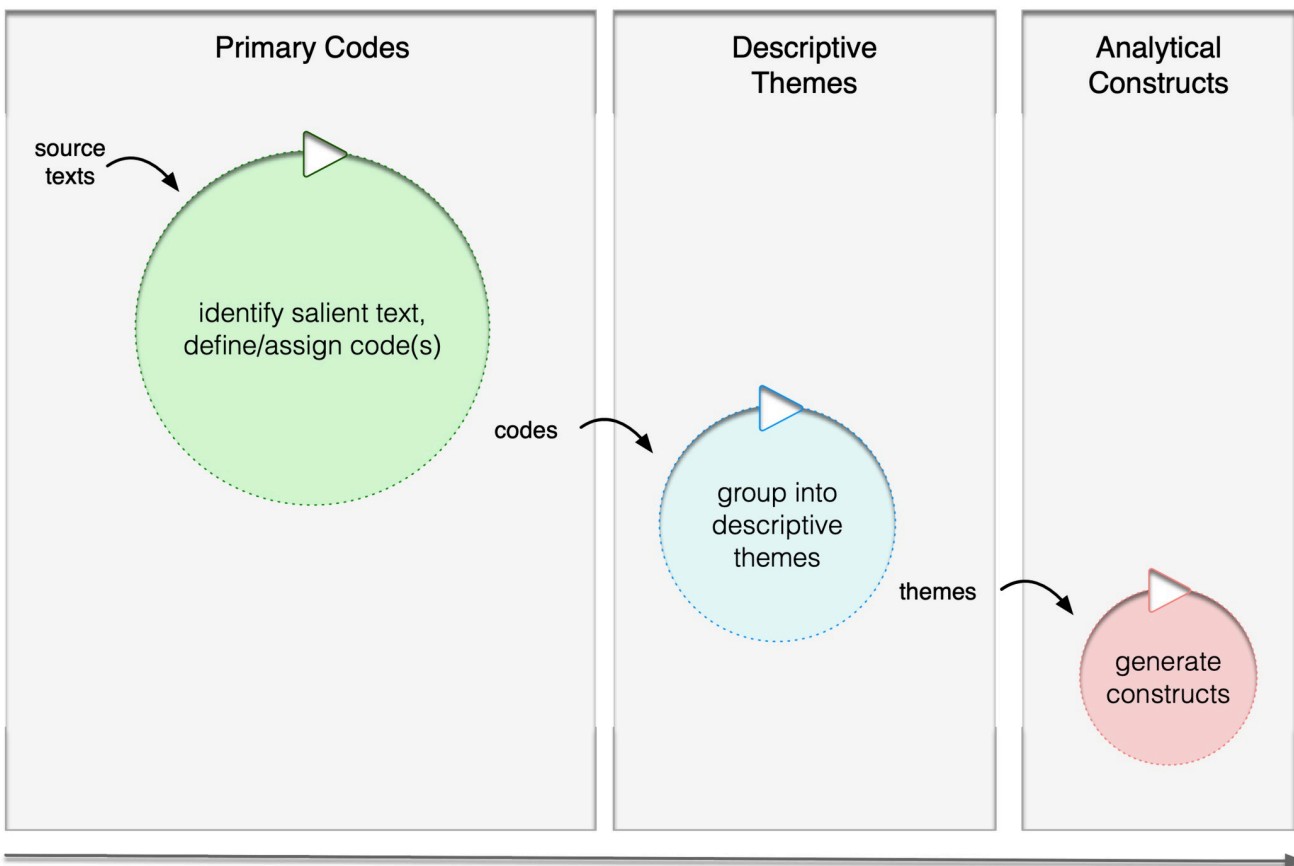

**Fig 2. Codes and themes to analytical constructs.**

[22,23]. Densities will be used to identify major and minor themes in the preliminary analysis. Higher-level analytical or theory-driven constructs will be generated from descriptive themes until all can be explained as factors influencing the use of PA sensors in stroke patients. These metrics will also be used to generate sunburst charts for major and minor themes.

## Results and discussion

A review of the literature revealed existing studies of PA sensor-based interventions more often had quantitative designs, with little qualitative focus specifically on stroke. By pooling existing qualitative research, the proposed study will identify major themes influencing PA sensor use, contributing in a novel way to a deeper understanding of the benefits and limitations of these technologies through the subjective experiential lens of older patients. To our knowledge this synthesis will be the first to explore themes relevant to the use of wearable PA sensors in older stroke patients and those with similar risk factors or deficits. The findings of this study may inform more effective device-based interventions in stroke prevention and rehabilitation. It is expected that these findings will also be relevant to technologists designing PA sensor devices, and therapists developing device-based interventions for older stroke cohorts. Results will be disseminated through open-access scholarly publications, including journal articles, as well as presentations at conferences and symposia.

## Supporting information

**S1 Checklist. PRISMA-P 2015 checklist.**
(DOC)

## Acknowledgments

The current protocol was adapted from a methodology described in a clinical research dissertation by PH. The preliminary coding structure and review process were jointly developed by IM and PH, based on the thematic synthesis methodology. Dr Phillis Lau from the Department of General Practice & Primary Care, University of Melbourne, reviewed the original dissertation and current protocol for methodological coherence. Review comments were also provided by Prof. Kathleen Gray from the Centre for Digital Transformation of Health, University of Melbourne. All authors have read and approved the final version of the manuscript.

## Author Contributions

**Conceptualization:** Paul T. Harris.

**Data curation:** Paul T. Harris.

**Formal analysis:** Paul T. Harris, Ingrid Maine.

**Methodology:** Paul T. Harris, Ingrid Maine.

**Project administration:** Paul T. Harris.

**Writing – original draft:** Paul T. Harris.

**Writing – review & editing:** Paul T. Harris, Ingrid Maine.

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
