## [Decision Letter · Decision Letter 0]

14 Nov 2023

PONE-D-23-26672Patient-perceived factors influencing physical activity sensor use in stroke prevention and rehabilitation: a thematic synthesis protocolPLOS ONE

Dear Dr. Harris,

Thank you for submitting your manuscript to PLOS ONE. After careful consideration, we feel that it has merit but does not fully meet PLOS ONE’s publication criteria as it currently stands. Therefore, we invite you to submit a revised version of the manuscript that addresses the points raised during the review process.

We look forward to receiving your revised manuscript.

Kind regards,

Daswin De Silva

Academic Editor

PLOS ONE

Journal Requirements:

Reviewers' comments:

Reviewer's Responses to Questions

**Comments to the Author**

1. Does the manuscript provide a valid rationale for the proposed study, with clearly identified and justified research questions?

Reviewer #1: Partly

Reviewer #2: Partly

2. Is the protocol technically sound and planned in a manner that will lead to a meaningful outcome and allow testing the stated hypotheses?

Reviewer #1: Yes

Reviewer #2: Partly

3. Is the methodology feasible and described in sufficient detail to allow the work to be replicable?

Reviewer #1: Yes

Reviewer #2: Yes

4. Have the authors described where all data underlying the findings will be made available when the study is complete?

Reviewer #1: Yes

Reviewer #2: Yes

5. Is the manuscript presented in an intelligible fashion and written in standard English?

Reviewer #1: Yes

Reviewer #2: Yes

6. Review Comments to the Author

You may also provide optional suggestions and comments to authors that they might find helpful in planning their study.

Reviewer #1: This study presents a thematic synthesis protocol. It is well written and strucutred. Following are some comments to improve:

1. How are the summaries created? Is it a manual process or automated, if automated indicate the methods and tools you propose to use

2. Recommend to indicate the use of inter-rater reliability measure for the screening process

3. Authors need to explain the coding framework in detail for the readers to understand the criterion used

4. Recommend to further strengthen the research gap in the introduction to highlight the importance of your study.

Reviewer #2: Line 45 - 52: please provide citations.

Line 67 - do you mean...efficacy in "Stroke prevention/management"?

Line 86 - is it healthcare intervention efficacy OR use of PA sensors in particular?

Line 91 - I question the time range. Why have the last 3 years of data been excluded? refer to mention of time period in Line 108 109 as well.

Line 93 - There is no rationale provided for selecting to focus only on older adults 50 years +

Line 137 -138 : What is the definition of "analytical richness" and is there a specific method or framework which will guide this process?

Line 148 - please provide a citation for "Axial coding" and perhaps explain in some detail, what this entails.

The research question/questions have not been clearly stated anywhere and is a significant gap.

Apart from the above, this protocol development indicates a good effort to follow accepted methods and guidelines on undertaking systematic reviews.

7. PLOS authors have the option to publish the peer review history of their article (what does this mean?). If published, this will include your full peer review and any attached files.

Reviewer #1: No

Reviewer #2: No

---

## [Author Response · Author response to Decision Letter 0]

17 Feb 2024

Regarding Comments to the Author (CTA):

CTA-1. The manuscript has been amended to explain the rationale for the study, including research questions in the final paragraph of the Introduction.

CTA-2. The data collection and analysis procedures have been described more fully in the revised manuscript. Data collection and analysis are also described in the Data Management Plan.

Regarding Reviewer #1 comments (R1C): 

R1C-1. The thematic synthesis process, well established in the qualitative research literature, will be recorded in reflexive notes generated by the two reviewers during this highly collaborative phase of the study. The summaries and preliminary analysis will be created in Microsoft Excel, based on coding density counts exported from NVivo. Summary measures will identify major and minor themes and these metrics will also be used to create sunburst charts in Excel for barrier and facilitator themes. The manuscript has been amended accordingly.

R1C-2. Regarding the recommendation to measure inter-rater reliability during screening, as there will be only two reviewers, we propose instead to calculate a simple agreement rate, and have updated the protocol to include this. Additionally, a pilot test will be performed, screening example article(s) against the inclusion and exclusion criteria to ensure mutual understanding between reviewers. Discrepancies in screening and coding will be resolved through discussion, or the engagement of a third party where consensus cannot be reached. Standard data extraction and screening forms will be used (Supplementary Materials, sections C & E). Detailed records of the screening process will be maintained, as described in the Supplementary Materials document.

R1C-3. The a priori coding framework (node structure) is illustrated in Fig 1 and described in the manuscript and Supplementary Materials.

Regarding Reviewer #2 comments (R2C): 

R2C-1. Included citations for stroke risk factors, and PA guidelines.

R2C-2. Updated to clarify that efficacy refers to stroke prevention and rehabilitation.

R2C-3. Updated to clarify that context is with respect to the efficacy of PA sensor-based interventions.

R2C-4. The bibliographic search was expanded to 31 December 2023, and the projected results availability shifted to June 2024.

R2C-5. Rationale for age range explained with citations in the second paragraph of the introduction.

R2C-6. “Analytical richness” defined, with citation.

R2C-7. “Axial coding” defined, with citation.

R2C-8. Research questions have been added to the last paragraph of the Introduction.

---

## [Decision Letter · Decision Letter 1]

27 Mar 2024

Patient-perceived factors influencing physical activity sensor use in stroke prevention and rehabilitation: a thematic synthesis protocol

PONE-D-23-26672R1

Dear Dr. Harris,

We’re pleased to inform you that your manuscript has been judged scientifically suitable for publication and will be formally accepted for publication once it meets all outstanding technical requirements.

Kind regards,

Daswin De Silva

Academic Editor

PLOS ONE

Additional Editor Comments (optional):

Reviewers' comments:

Reviewer's Responses to Questions

**Comments to the Author**

1. Does the manuscript provide a valid rationale for the proposed study, with clearly identified and justified research questions?

Reviewer #1: Yes

2. Is the protocol technically sound and planned in a manner that will lead to a meaningful outcome and allow testing the stated hypotheses?

Reviewer #1: Yes

3. Is the methodology feasible and described in sufficient detail to allow the work to be replicable?

Reviewer #1: Yes

4. Have the authors described where all data underlying the findings will be made available when the study is complete?

Reviewer #1: Yes

5. Is the manuscript presented in an intelligible fashion and written in standard English?

Reviewer #1: Yes

6. Review Comments to the Author

You may also provide optional suggestions and comments to authors that they might find helpful in planning their study.

Reviewer #1: The authors have addressed the previous comments to strengthen the review process and include additional details for the methodology. The manuscript is well written and presented in a clearer manner in the revision. No further comments, provided that citations/ formatting adhere to the journal guidelines.

7. PLOS authors have the option to publish the peer review history of their article (what does this mean?). If published, this will include your full peer review and any attached files.

Reviewer #1: No
